# Turning Speech Language Models into Multilingual Listeners

## Abstract

Speech Language Models (SLMs) that understand spoken language questions and commands support only a few high-resource languages, limiting access to modern technology for millions of speakers worldwide. This gap in language coverage stems from the scarcity of multilingual speech-language instruction-tuning datasets. To address this issue, we present MULTISPEECHQA, a large-scale, synthetically generated and human-verified dataset comprising 9200 hours of more than 10.8 million spoken question-answer pairs in 23 typologically diverse languages, designed to improve the multilingual instruction-following capabilities of SLMs. Using MULTISPEECHQA, we also introduce MULTISPEECH-BENCH, a multi-task benchmark to evaluate SLM performance across 23 languages. We compare the performance of a strong cascading system to three leading open-weight SLMs on MULTISPEECH-BENCH and find that the cascading system outperforms all existing open-weight SLMs. We then demonstrate the effectiveness of MULTISPEECHQA by fine-tuning the best-performing open-weight SLM, Qwen 2.5-Omni, on our dataset, which substantially improves its performance and establishes new state-of-the-art results for open-weight models on our benchmark. Our findings show that high-quality synthetic datasets offer a scalable solution to improving the multilingual capabilities of SLMs, extending the benefits of natural spoken interactions to a wider range of languages.

## 1 Introduction

Speech Language Models (SLMs) often combine a pretrained speech encoder with a pretrained Large Language Model (LLM), using a modality adapter module to map the output of the speech encoder into the language model input space to perform various speech and language processing tasks (Arora et al., 2025). These models are trained with instruction tuning data to align the speech encoder and LLM, and allow for natural spoken interactions.

SLMs have many advantages over alternatives like the popular multitask speech model Whisper (Radford et al., 2023), including allowing natural language instructions for speech tasks, doing question answering out-of-the-box and enabling zero-shot performance in a variety of traditional speech processing tasks, such as emotion recognition, audio captioning or audio-based storytelling. However, the open-weight SLMs that exist today are primarily developed for English and a few other high-resource languages (Zhang et al., 2023; Chu et al., 2024b; Fang et al., 2024; Tang et al., 2024; Abouelenin et al., 2025). This limits access to the state-of-the-art speech-language technology for many speakers worldwide.

The most critical challenge in developing multilingual SLMs is the scarcity of multilingual speech-language instruction-tuning datasets. While there has been significant progress on curating such multilingual data for text-only models (Singh et al., 2024; Üstün et al., 2024), and vision-language models (Dash et al., 2025; Yue et al., 2025), the intersection of speech and language remains severely limited.

There are several benchmarks that have been introduced to measure speech language model capabilities. These include speech and audio understanding of AudioBench (Wang et al., 2025), spoken language understanding with SLUE (Shon et al., 2022) and safety, bias and fairness evaluation with AHELM (Lee et al., 2025) and AIR-Bench (Yang et al., 2024). The existing evaluation benchmarks

for SLMs also suffer from the lack of language coverage, in particular, for open-ended generative instruction following.

To address this gap, we present MULTISPEECHQA, a large-scale multilingual spoken question-answering (SQA) dataset comprising of 10.8 million instructions and 9200 hours of synthetically generated and human-verified speech data in 23 typologically diverse languages. MULTISPEECHQA consists of open-ended question-answer pairs from variety of tasks, and data sources designed to foster instruction following capabilities for SLMs in 23 languages. We combine MULTISPEECHQA with CommonVoice (Ardila et al., 2020) automatic speech recognition (ASR) data and CovST-2 (Wang et al., 2021) automatic speech translation (AST) data to create MULTISPEECH-BENCH, providing a multi-task evaluation suite of these models in 23 languages.

Our main contributions are as follows:

1. We validate the hypothesis that automated synthetic data generation can provide sufficiently good instruction-tuning data to enable effective post-training of SLMs for many languages, provided only that adequate machine translation (MT) and speech synthesis (TTS) systems exist for those languages.

2. We provide MULTISPEECHQA, a **multi**lingual speech-language instruction fine-tuning dataset that consists of over 10.8 million **spoken question-answer** pairs in 23 languages, where multilingual samples are generated by using translation and speech synthesis, comprising 9200 hours in total.

3. We develop MULTISPEECH-BENCH, a **multi**lingual, **multi**task **speech** processing benchmark, facilitating evaluation of speech recognition, speech translation and spoken question answering in 23 languages.

4. Validating the effectiveness of our dataset, we finetune Qwen2.5-Omni on MULTISPEECHQA and show that it achieves state-of-the-art performance among open-weight models on MULTISPEECH-BENCH, particularly outperforming Qwen2.5-Omni with 60% win-rate across 23 languages.

By releasing our dataset and model weights, we aim to extend the benefits of modern speech technology to speakers of diverse languages worldwide. Our dataset, benchmark and models are publicly available.

## 2 RELATED WORK

**Speech Language Models.** SLMs can be broadly categorized into three architectural approaches: (1) models of speech distribution; (2) models of joint speech-text distribution, and (3) models combining pre-trained text LLMs with speech encoders (Arora et al., 2025). The third approach leverages the instruction-following capabilities learned by the text LLM and typically requires less training data, enabling strong few-shot or zero-shot performance on a variety of multimodal tasks (Chen et al., 2024). Many state-of-the-art models adopt this approach, including proprietary models, such as Gemini 2.5 (Comanici et al., 2025) and GPT-4o (OpenAI et al., 2024), as well as notable open-source models, such as Phi-4-Multimodal (Abouelenin et al., 2025), SALMONN (Tang et al., 2024), and Qwen2Audio (Chu et al., 2024b).

Comparing open-source models reveals limited multilingual support. SALMONN is primarily trained on English data, while Phi-4-Multimodal and Qwen2Audio support only eight languages. Both SALMONN and Qwen2Audio leverage a Whisper-based encoder (Radford et al., 2023), which is aligned with an LLM backbone, suggesting potential for broader language coverage that remains largely unexplored. Our work substantially extends the language coverage of these models by providing support for 23 languages with a comprehensive evaluation.

**Multilingual SQA Datasets.** Multilingual SQA datasets are scare, limiting the development of truly multilingual SLMs. Existing multilingual speech benchmarks and datasets primarily target traditional tasks, rather than open-ended SQA. For example, ASR and AST dataset FLEURS (Conneau et al., 2023) and ASR dataset ML-SUPERB 2.0 (Shi et al., 2024) cover 102 and 143 languages, respectively. For SQA specifically, Voice Assistant 400K (Xie & Wu, 2024) offers diverse question-answer pairs but only in English. Additionally, recent work shows that high-quality synthetic speech

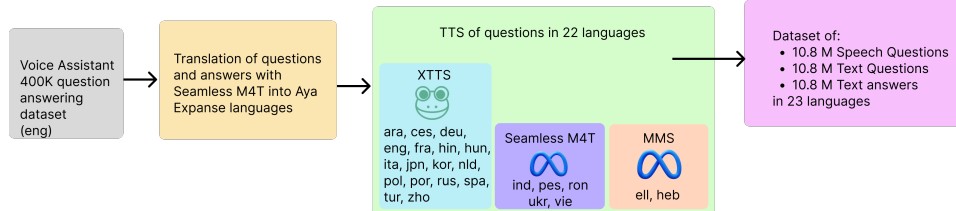

Figure 1: Summary of the dataset creation process. We translate the questions and answers of the Voice Assistant 400K dataset into each of the Aya Expanse languages (in Table 2) with Seam­lessM4T, synthesise the questions with XTTS for the languages covered by the model and Seam­lessM4T for the languages not covered by XTTS, leaving us with roughly 10.8 million text ques­tions, speech questions and text answers.

can effectively augment limited real data. Phi-4-Multimodal demonstrated strong performance in SQA tasks using synthetic speech from translations. Our training approach leverages this insight, while substantially expanding the language coverage with MULTISPEECHQA.

## 3  DATASET CREATION

Figure 1 presents our two-stage process to create MULTISPEECHQA. We build upon the English Voice Assistant 400K (VA 400K; Xie & Wu, 2024) dataset, which consists of synthesised speech from text-only instruction-completion pairs. These instruction-completion pairs are sourced from multiple datasets, as detailed in Table 1. We extend VA 400k to 22 additional languages covered by the Aya Expanse 8B (Dang et al., 2024) model through translation and synthesis, which has been shown to be effective in past work (Abouelenin et al., 2025). We summarise the languages in our dataset in Table 2.

Table 1: Splits in our dataset with number of instruction-completion pairs per language.

| Dataset | Number of pairs per language |
|---|---|
| Trivia (Multi-choice, 17K) (Mihai, 2024c) | 16,528 |
| Trivia (Single-choice, 20K) (Mihai, 2024c) | 16,529 |
| QA Assistant V1 (7K) (Mihai, 2024a) | 5,769 |
| QA Assistant V2 (20K) (Mihai, 2024b) | 16,008 |
| Alpaca GPT-4 (EN, 55K) (Peng et al., 2023) | 31,293 |
| Identity (Xie & Wu, 2024) | 4,306 |
| RLHF (Bai et al., 2022) | 379,621 |
| **Total** | **470,054** |

### 3.1  TRANSLATION AND SYNTHESIS

For translation, we use Seamless M4T v2 Large (Seamless Communication et al., 2023) to translate instruction-completion pairs from English into the 22 target languages languages. This model was chosen as it is publicly available, free to use, and it achieves stronger performance compared to other models of similar size, such as NLLB (Team et al., 2022), in our preliminary experiments.

For speech synthesis, we use different models based on language support. We use XTTS (Casanova et al., 2024) for 15 languages, as we found its audio quality to be superior to other models in our preliminary evaluations. For the remaining seven languages not supported by XTTS, we use Seam­less M4T v2 Large and language-specific MMS text-to-speech (TTS) models (Pratap et al., 2024). To improve speaker diversity in the training data, which is important for achieving robust perfor­mance (e.g., see Jia et al., 2018), we leverage XTTS's voice cloning capability with short LibriVox (McGuire, 2005) clips of perceived male and female speakers. For each language supported by XTTS, we randomly select a voice, which might be male or female, from all the LibriVox clips dur-

Table 2: Languages in MULTISPEECHQA with their language families, ISO 639-3 codes, TTS model used, and human evaluation scores for naturalness and content understanding.

| Language | Language Family | ISO 639-1 | TTS Model | Naturalness | Content Understood |
|---|---|---|---|---|---|
| Arabic | Afro-Asiatic (Semitic) | ar | XTTS | 2.8 | 3.7 |
| Chinese (Simplified) | Sino-Tibetan (Sinitic) | zh | XTTS | 2.8 | 4.8 |
| Czech | Indo-European (Slavic, West) | cs | XTTS | – | – |
| Dutch | Indo-European (Germanic, West) | nl | XTTS | 3.1 | 4.4 |
| English | Indo-European (Germanic, West) | en | – | – | – |
| French | Indo-European (Romance) | fr | XTTS | 3.6 | 4.3 |
| German | Indo-European (Germanic, West) | de | XTTS | 3.0 | 4.4 |
| Greek | Indo-European (Hellenic) | el | MMS | 2.4 | 4.2 |
| Hebrew | Afro-Asiatic (Semitic) | he | MMS | 2.1 | 2.4 |
| Hindi | Indo-European (Indo-Aryan) | hi | XTTS | 3.4 | 3.5 |
| Indonesian | Austronesian (Malayo-Polynesian) | in | XTTS | 3.0 | 4.1 |
| Italian | Indo-European (Romance) | it | XTTS | 3.5 | 4.5 |
| Japanese | Japonic | ja | XTTS | 3.0 | 2.9 |
| Korean | Koreanic | ko | XTTS | 2.3 | 4.2 |
| Farsi | Indo-European (Iranian) | fa | Seamless | 2.5 | 4.2 |
| Polish | Indo-European (Slavic, West) | pl | XTTS | 4.0 | 4.4 |
| Portuguese | Indo-European (Romance) | pt | XTTS | 3.7 | 4.6 |
| Romanian | Indo-European (Romance) | ro | Seamless | 1.8 | 4.0 |
| Russian | Indo-European (Slavic, East) | ru | XTTS | 3.7 | 4.6 |
| Spanish | Indo-European (Romance) | es | XTTS | 3.6 | 4.9 |
| Turkish | Turkic (Oghuz) | tr | XTTS | 3.4 | 4.3 |
| Ukrainian | Indo-European (Slavic, East) | uk | Seamless | 2.5 | 4.6 |
| Vietnamese | Austroasiatic (Vietic) | vi | Seamless | 3.2 | 2.8 |

ing synthesis, resulting in 37 different voices across the dataset. Table 2 shows the model assignment per language.

## 3.2 HUMAN EVALUATION

To measure both the quality of the translations and synthesised speech, we conduct a human evaluation for our dataset. We sample 20 instruction-completion pairs for each language from our dataset, and ask native speakers of each language to evaluate both the naturalness of the speech and the amount of content they have understood on a 5-point scale (more details in Appendix A). We ensure that each language's examples were reviewed by at least two native speakers, except for Czech for which we could not obtain any ratings.

As shown in Table 2, scores for the perceived naturalness of the speech range from 1.8 to 4.0, and the scores for content understanding range from 2.4 to 4.9. Unsurprisingly, the average score for naturalness (3.0) falls behind the content understanding (4.1), as the speech synthesis models often struggle to generate the highest quality natural sounds in many languages (Casanova et al., 2024; Pratap et al., 2024).

Comparing the models used for speech synthesis, XTTS shows better performance than SeamlessM4T and language-specific MMS TTS models, achieving an averaged score of 3.3 and 4.2 in 15 languages for naturalness and the amount of content understood, respectively. Results for the language-specific MMS TTS models are 2.25 and 3.3 averaged across two languages, and SeamlessM4T are 2.5 and 3.9. Note that the languages that use MMS TTS models where XTTS does not have language coverage, are lower-resource languages such as Farsi and Greek. These results show that our dataset is adequate for multilingual instruction finetuning, while further improvements will most strongly depend on improving TTS quality.

## 3.3 MULTISPEECH-BENCH FOR MULTILINGUAL AND MULTITASK EVALUATION

We split the MULTISPEECHQA dataset into train, development and test sets. We randomly sample SQA pairs of the different subsets with the same dataset distribution as VA 400K, resulting in a development set of 2000 SQA pairs and a test set of 1000 SQA pairs. To avoid speaker overlap, we select different speakers for the train and test set where possible. All remaining data belongs to the train set.

We create MULTISPEECH-BENCH from a subset of our test split, sampling the same 200 SQA pairs per language. To ensure the quality of this evaluation dataset, we collect human annotations

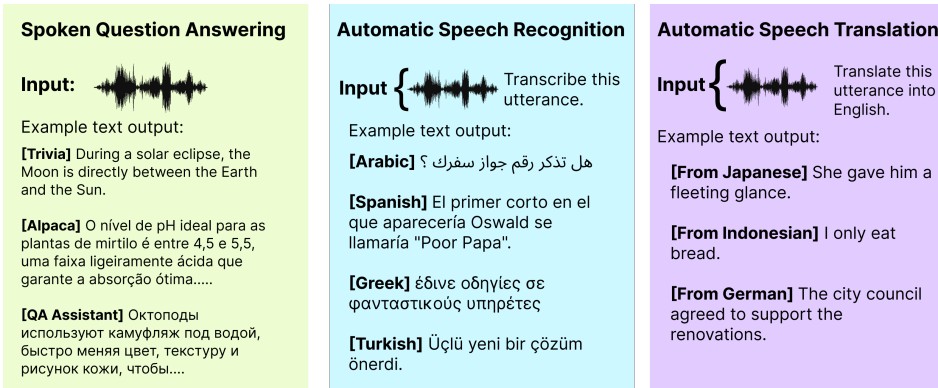

Figure 2: MULTISPEECH-BENCH covers three tasks: (1) Spoken Question Answering (SQA), where models are prompted with speech only (no text prompt); (2) Automatic Speech Translation (AST) from speech to English, using CoVoST-2 (X to En) for languages that overlap with the 23 languages in our dataset; and (3) Automatic Speech Recognition (ASR) on CommonVoice for each of the 23 supported languages.

on all 200 SQA pairs using Prolific, asking language experts to review and correct the translations where necessary. Overall, 72% of translations required editing with language-specific correction rates ranging from 43% (Turkish) to 86% (Chinese). This manually verified subset is combined with existing test from CommonVoice (ASR) and CoVST-2 (ASR) for matching languages, creating our multilingual, multitask benchmark.

## 4    EVALUATION ON OPEN-WEIGHT MODELS

To establish the performance of current multilingual SLMs, we evaluate leading open-weight SLMs on MULTISPEECH-BENCH and compare it against a strong cascading system baseline. This evaluation quantifies the performance gap between languages and establishes baselines for measuring SQA performance improvements from training with MULTISPEECHQA. For the SQA portion of MULTISPEECH-BENCH, we adopt pairwise preference evaluations using LLM-as-a-judge, following recent work involving open-ended multilingual generation (Üstün et al., 2024). This approach allows for consistent evaluation across our 23 languages, and is more cost-efficient than recruiting human annotators for each language. We use the multilingual Command-A (Cohere et al., 2025) model as our LLM-as-a-judge, which supports the 23 languages in our datasets (more details in Appendix A.5). To check for calibration across LLMs, we also use GPT-4o as an LLM judge.

**Cascading System Baseline**   While end-to-end models that process speech directly have architectural advantages (e.g., preserving acoustic information), we include a strong cascading system baseline by first transcribing the speech with Whisper Large v3 (Radford et al., 2023), then prompting Aya Expanse 8B (Dang et al., 2024) with the transcription.

Whisper is a leading multilingual model that supports over 100 languages and is trained to do ASR and AST into English. Aya Expanse 8B is a language model trained to respond to questions in all 23 languages in MULTISPEECHQA. Such cascading baselines perform often on par or exceed the performance of SLMs on some spoken language processing tasks (Chen et al., 2024). Our benchmark can help us learn whether this is true for our three tasks, even though non-cascading SLMs have many other advantages – for example, they are the only choice for performing speech-native tasks like spoken emotion detection or speaker identification.

**Open-Weight Models**   We evaluate three leading open-weight SLMs that represent different training approaches and cover different languages:

1. **Qwen2-Audio** (Chu et al., 2024a): A multimodal model that combines an audio encoder initialized from Whisper Large v3 (Radford et al., 2023) with a QwenLM 7B decoder (Chu

et al., 2024b). The modality adapter is a multi-layer perceptron. The authors do not specify full language support, but model performance is reported on English, French, and Chinese.

2. **Qwen2.5-Omni** (Xu et al., 2025): A multimodal model incorporating speech and vision modalities into the Qwen2.5 language model. The processing of multimodal inputs and text generation happens in the 'Thinker' part of the model. For speech, it uses an encoder that is initialized with Whisper Large v3, and a multi-layer perceptron as the modality adapter. The languages supported by the model are not explicitly stated.

3. **Phi-4 Multimodal** (Abouelenin et al., 2025): A multimodal model incorporating speech and vision modalities into the Phi-4 language model. For speech, it uses a conformer model, which is trained on a proprietary dataset. The modality adapter is a multi-layer perceptron. The model supports English, Chinese, German, French, Italian, Japanese, Spanish, and Portuguese audio input.

**Commercial SLMs**   We evaluate two leading commercial SLMs:

1. **GPT-Audio:** GPT-Audio is OpenAI's speech-enabled variant of the GPT family, designed for real-time multimodal interaction. It supports speech recognition, speech-to-text reasoning, and text-to-speech generation within a unified model.

2. **Gemini 2.5 Flash Lite:** Gemini 2.5 Flash Lite is a compact member of Google's Gemini 2.5 series. The model provides robust automatic speech recognition and basic audio-event understanding.

## 4.1 RESULTS

With Whisper combined with Aya Expanse 8B as the baseline, Figure 4 shows win rates on the SQA portion of MULTISPEECH-BENCH for each open-weight model tested against it. We find that Qwen2.5-Omni outperforms all other open-weight models, and our analysis of language-specific win rates reveals that SLMs perform better on the languages they explicitly support (details in Appendix A.1). Qwen2-Audio and Phi-4-Multimodal are competitive with the baseline in languages that the models are trained on, but it is clear that they are outperformed by the cascading system baseline, likely due to the strength of the individual ASR and language models on their specific tasks and the lack of catastrophic forgetting that can occur during instruction tuning of the models to enable multimodal processing.

In Table 3, we show the performance of the baseline and SLM models on ASR and AST, measuring ASR performance using the word error rate (WER; character error rate (CER) for Chinese (zh) and Japanese (ja)) and AST performance with BLEU (Papineni et al., 2002) and chrF (Popović, 2015). Qwen2.5-Omni shows the strongest performance among the evaluated SLM models (average ASR error rate of 49.7; average BLEU of 22.7; average chrF of 46.6), outperforming the baseline on AST. The per-language results are mixed for both tasks, but generally models perform strongest on the languages seen during training.

## 5 FINETUNING WITH MULTISPEECHQA

The open-weight model results show the need for further model improvement. We take the best-performing open-weight SLM, Qwen2.5-Omni and finetune it on MULTISPEECHQA. We perform LoRA finetuning (Hu et al., 2022) on all linear modules in each transformer layer, using a rank of 32. We train for a fixed number of steps, equalling roughly 3 epochs of the data. We then evaluate its performance on MULTISPEECH-BENCH.

## 5.1 RESULTS

Figure 5 shows the win rates of Qwen2.5-Omni finetuned with MULTISPEECHQA against the non-finetuned Qwen2.5-Omni model. We find that parameter efficient finetuning improves SQA performance substantially. The finetuned model wins the majority of the time, struggling with languages such as Hebrew, Greek and Farsi, where the judgements tie 48.0% of the time on average. When considering all languages, our finetuned model wins 60.6% of the time on average. This finetuned

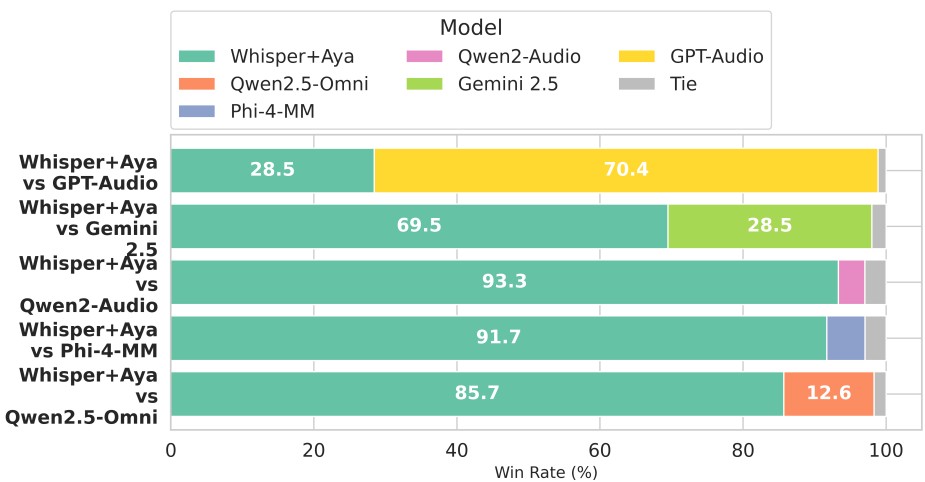

Figure 3: Win rates on MULTISPEECH-BENCH averaged across languages for the open-weight SLMs against the baseline cascading system of Whisper combined with Aya Expanse 8B using the Command-A LLM-as-a-Judge. Bars show % wins for each model and % ties (gray).

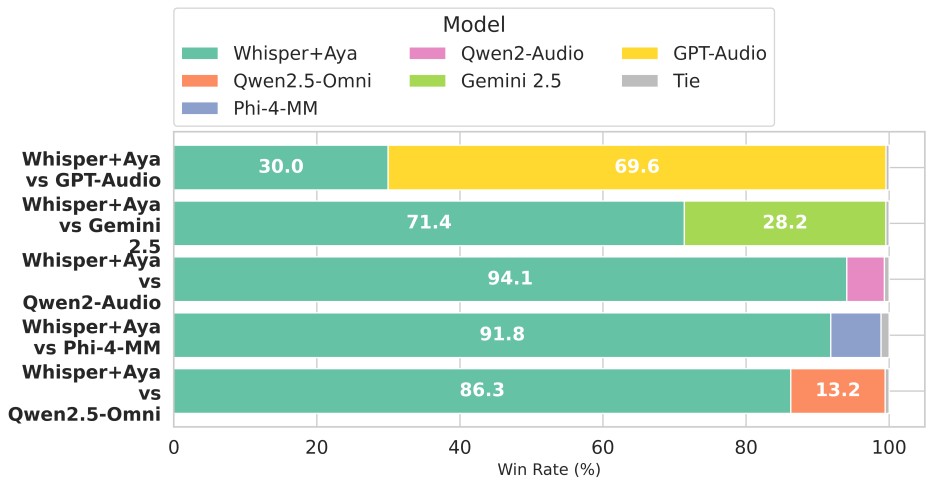

Figure 4: Win rates on MULTISPEECH-BENCH averaged across languages for the open-weight SLMs against the baseline cascading system of Whisper combined with Aya Expanse 8B using the GPT-4o LLM-as-a-Judge. Bars show % wins for each model and % ties (gray).

Table 3: Speech Recognition (ASR) and Speech Translation (AST) performance across models and languages. The baseline results are for the cascaded Whisper and Aya-Expanse 8B model. We compare the baseline to Qwen2-Audio (Q2-Audio), Phi-4 Multimodal (Phi-4 MM), and Qwen2.5-Omni (Q2.5-Omni). We report CER for Chinese (zh) and Japanese (ja), and report chrF in addition to BLEU for AST performance.

| Lang. | ASR (WER %) ↓ | | | | AST | | | |
|---|---|---|---|---|---|---|---|---|
| | Baseline | Q2-A | Phi4-MM | Q2.5O | Baseline BLEU/chrF | Q2-A BLEU/chrF | Phi4-MM BLEU/chrF | Q2.5O BLEU/chrF |
| ar | 14.0 | 118.1 | 146.1 | 45.7 | 34.2/54.7 | 7.3/30.9 | 0.1/11.9 | 30.6/53.8 |
| cs | 26.0 | 117.4 | 115.2 | 100.3 | –/– | –/– | –/– | –/– |
| de | 9.4 | 33.3 | 7.0 | 7.1 | –/– | –/– | –/– | –/– |
| el | 21.8 | 118.3 | 114.1 | 108.0 | –/– | –/– | –/– | –/– |
| en | 3.2 | 34.6 | 13.9 | 13.8 | –/– | –/– | –/– | –/– |
| es | 7.7 | 18.1 | 4.9 | 4.9 | –/– | –/– | –/– | –/– |
| fa | 38.0 | 128.7 | 133.0 | 109.1 | –/– | –/– | –/– | –/– |
| fr | 9.0 | 34.1 | 10.1 | 10.9 | –/– | –/– | –/– | –/– |
| he | 40.6 | 128.4 | 447.8 | 122.9 | –/– | –/– | –/– | –/– |
| hi | 30.8 | 123.2 | 105.3 | 68.9 | –/– | –/– | –/– | –/– |
| id | 34.9 | 71.8 | 125.1 | 14.0 | 36.1/53.3 | 6.6/29.8 | 0.2/15.0 | 37.0/59.0 |
| it | 5.0 | 21.7 | 5.1 | 6.5 | –/– | –/– | –/– | –/– |
| ja | 15.8 | 66.1 | 78.6 | 75.9 | 10.4/23.4 | 11.3/38.5 | 19.7/46.6 | 17.8/41.5 |
| ko | 20.9 | 61.4 | 144.4 | 23.0 | –/– | –/– | –/– | –/– |
| nl | 9.5 | 90.8 | 101.5 | 14.0 | –/– | –/– | –/– | –/– |
| pl | 7.5 | 110.8 | 118.6 | 94.7 | –/– | –/– | –/– | –/– |
| pt | 6.7 | 28.5 | 7.4 | 9.6 | –/– | –/– | –/– | –/– |
| ro | 15.1 | 114.6 | 106.1 | 89.3 | –/– | –/– | –/– | –/– |
| ru | 17.1 | 57.4 | 123.9 | 9.3 | –/– | –/– | –/– | –/– |
| tr | 11.4 | 114.6 | 131.9 | 74.5 | 20.0/40.8 | 0.6/19.3 | 0.1/15.4 | 5.5/27.0 |
| uk | 18.7 | 107.0 | 118.8 | 82.6 | –/– | –/– | –/– | –/– |
| vi | 18.0 | 110.5 | 104.2 | 50.9 | –/– | –/– | –/– | –/– |
| zh | 28.9 | 93.9 | 7.9 | 6.2 | 4.5/16.9 | 15.6/45.8 | 8.9/39.7 | 22.7/51.2 |
| Ave. | 17.8 | 82.8 | 98.7 | 49.7 | 21.0/37.8 | 8.3/32.9 | 5.8/25.7 | 22.7/46.6 |

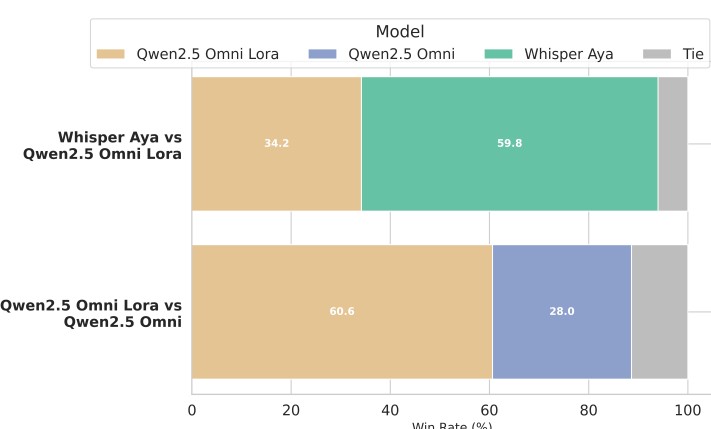

Figure 5: Win rates on MULTISPEECH-BENCH averaged across languages for the Qwen2.5-Omni and the Qwen2.5-Omni model finetuned on MULTISPEECHQA. Bars show % wins for each model and % ties (gray).

model also performs best on SQA against the cascading baseline, leading to state-of-the-art SLM performance on the SQA portion of MULTISPEECH-BENCH.

Comparing the ASR and AST performance of Qwen2.5-Omni and Qwen2.5-Omni finetuned on MULTISPEECHQA, we find that the average ASR performance remains stable across languages

(per-language results shown in Table 4 in Appendix A.4). The average WER increases marginally from 49.7 for the non-finetuned model to 50.4 for the finetuned model. On AST, the performance of the finetuned model is similarly comparable, as shown by the slightly lower BLEU score of 21.0 compared to 22.7 for the non-finetuned model. Overall, we find that MULTISPEECHQA finetuning substantially improves spoken SQA, while leaving performance on core ASR and ASR capabilities effectively unchanged.

# 6 HOW DO TRAINING DATA MIXTURES AFFECT SPEECH LANGUAGE MODEL PERFORMANCE?

In Section 5, we show that MULTISPEECHQA improves the SQA performance of Qwen2.5-Omni. These results motivate a controlled study of data composition for multilingual SLMs. Specifically, most existing SLMs, including Qwen2.5-Omni, are trained on undisclosed data mixtures, making it impossible to understand whether performance differences across languages arise from the model capacity being spread across many languages or from insufficient task diversity. We therefore ask two questions: (1) Does training with fewer languages lead to better performance?; and (2) Does adding AST data improve model performance?

To answer these questions, we train models from scratch using the SALMONN (Tang et al., 2024) architecture, whose training code is publicly available. Specifically, in our setup, we use Whisper as the speech encoder and Aya Expanse 8B as the language model. The window-level Q-Former uses an mBERT text encoder. We choose this setup, because the Whisper encoder produces stable multilingual speech features, and the window-level Q-Former allows us to leverage a pretrained text encoder to more efficiently learn intermediary representations.

As Whisper is trained to translate speech data of many of the Aya's languages into English, we hypothesise that adding AST data increases the task diversity in the training mixture and hence could lead a better downstream performance. For this ablation, we set a threshold of 20% for the speech translation data and use mixed batches for more robust multi-task instruction-tuning.

**Training details** We train our models in two stages: (1) multimodal alignment with ASR data, followed by (2) multitask training with SQA and AST data. Following the SALMONN training setup, we train the window-level Q-Former and LoRA adapters and keep the speech encoder and language model frozen. Although the model architecture allows us to append a text prompt to the speech input, we train the model without any additional text prompt with our question answering examples to enable the question-answering capability from the spoken questions alone. Further details on how we train the models are in Appendix A.6 and hyperparamter details are in Table 5.

In total, we train four models: (1) ALL+AST: a model trained on all of Aya's 23 languages with CoVoST-2 AST data; (2) ALL: a model trained on all of Aya's 23 languages; (3) TEN: a model trained with ten selected languages (English, French, Dutch, Turkish, German, Arabic, Spanish, Russian, Indonesian, and Polish); and (4) TEN+AST: a model trained with ten selected languages with CoVoST-2 AST data.

## 6.1 RESULTS

**Does training with fewer languages lead to better performance?** Figure 6 summarises the difference in SQA performance of the models trained with 10 languages (TEN/TEN+AST) and 23 languages (ALL/ALL+AST). We see that the model trained with 23 languages results in a better win rate overall, suggesting that we do not experience capacity dilution at 23 languages, and adding more languages in training leads to better performance across languages. This could be due to the fact that we start with a pretrained speech encoder and a pretrained language model, meaning that we already have the question-answering capabilities present in the LM and the SLM training is primarily learning how to project the speech encoder output into the LM space.

**Does adding AST performance improve the models?** We evaluate whether including speech translation data improves performance by testing on CoVoST-2 languages (both X to En and En to X translation directions) that overlap with the 23 languages in our dataset. We find that models trained with AST data win 46.8% of the time against those without, indicating that additional AST data alone does not lead to a consistent improvement. This result is likely due to two factors: (1)

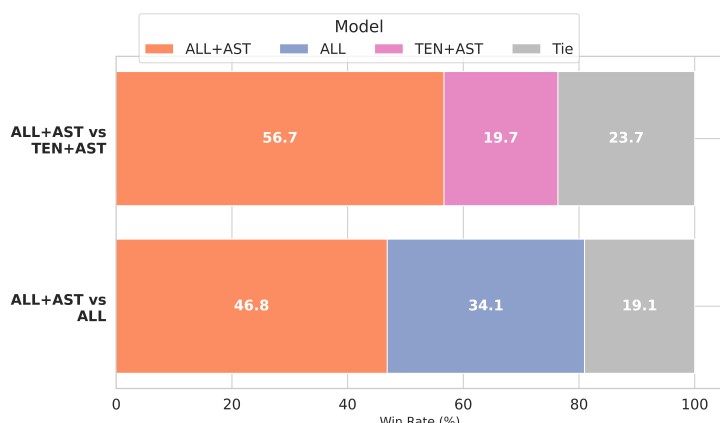

Figure 6: Win rates on MULTISPEECH-BENCH averaged across languages for the from-scratch SALMONN models. ALL+AST versus TEN+AST is shown at the top and versus ALL at the bottom. Bars show % wins for each model and % ties (gray).

the AST data comprises 20% of the training mixture, which is potentially too small to produce a measurable effect; and (2) Whisper already supports speech translation into English, so adding a small amount of AST instruction-tuning data provides only limited additional supervision.

## 7 CONCLUSION

In this paper, we address the lack of multilingual instruction-tuning data for SLMs by presenting MULTISPEECHQA, a synthetic, human-verified dataset of more than 10.8 million instructions and 9200 hours of spoken question-answering data in 23 languages. We also introduce MULTISPEECH-BENCH, a human-verified, multilingual and multitask benchmark to evaluate SLMs on SQA, ASR and AST. Using MULTISPEECH-BENCH, we establish the strong performance of Qwen2.5-Omni among the open-weight models we evaluate, and demonstrate the effectiveness of finetuning this model on MULTISPEECHQA, leading to state-of-the art performance on the SQA portion of MULTISPEECH-BENCH. These findings validate that automated, synthetic pipelines provide sufficient instruction-tuning data for effective post-training of SLMs across many languages.

## 8 LIMITATIONS

We present a synthetically generated dataset, which for several languages suggests the possibility of errors in the machine translation, which could lead to unnatural or possibly incorrect question prompts in our dataset. The quality of the generated speech is at the limit of the speech synthesis models, so we ensured that the content could be understood by native speakers for each language. Synthetic generation of speech means we have a limited number of voices despite the vast amounts of data in our dataset.

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

# A   APPENDIX

In these appendices, we provide additional experimental details and results. Appendix A.1 reports win rates of open-weight SLMs against the baseline. Appendix A.2 then describes the human evaluation procedure for synthesized questions. We further examine model performance in Appendix A.3, which presents win rates of Qwen2.5-omni against its fine-tuned variant trained on MULTISPEECHQA, followed by ASR and AST results for Qwen2.5-omni models in Appendix A.4. The LLM-as-a-Judge prompt used in our evaluations is provided in Appendix A.5. Finally, Appendix A.6 contains comprehensive training details.

## A.1   WIN RATES OF OPEN-WEIGHT MODELS AGAINST THE WHISPER + AYA BASELINE

We show language-wise win-rate breakdowns for each of the SLMs against the baseline model in Figures 7, 8 and 9.

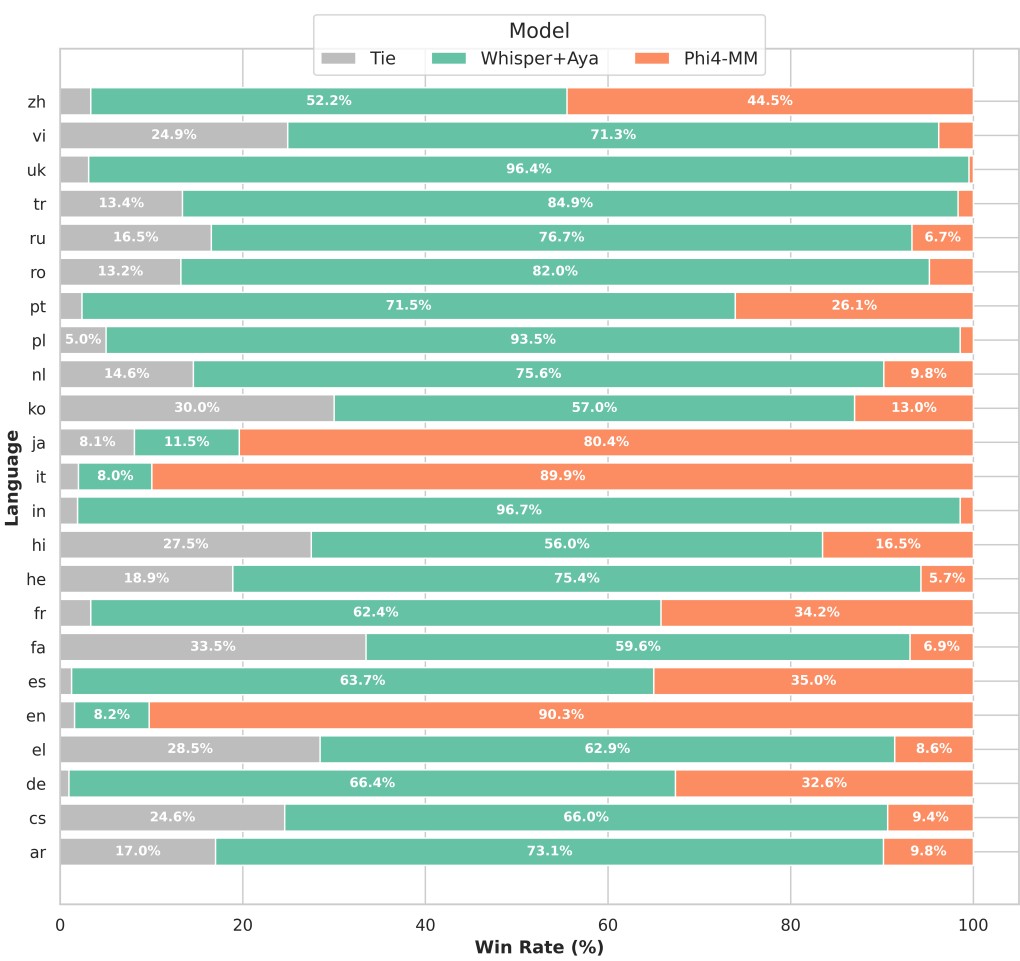

Figure 7: Win rates comparison: Whisper + Aya vs. Phi-4-Multimodal. Whisper + Aya outperforms Phi-4-Multimodal on most languages, with the exception of a subset on which Phi-4-Multimodal is trained (i.e., English, Italian, and Japanese).

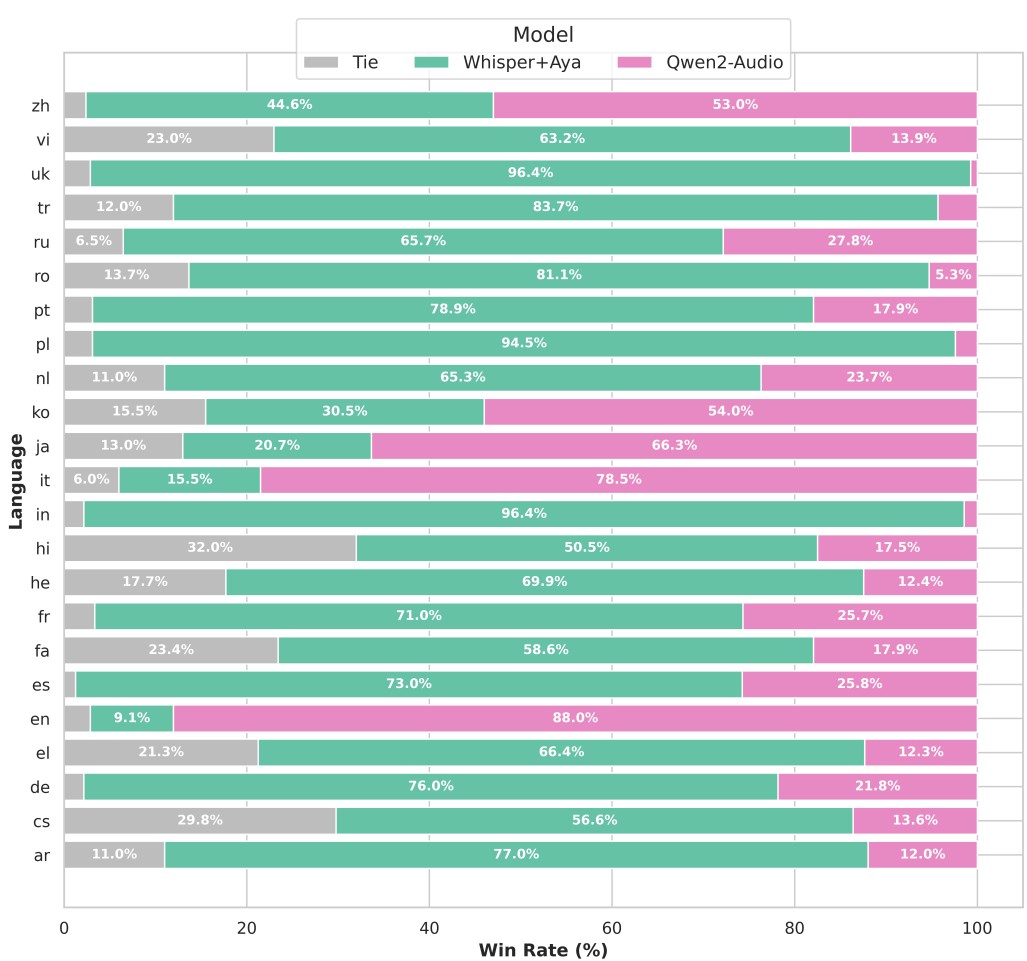

Figure 8: Win rates comparison: Whisper + Aya vs. Qwen2-Audio. Whisper + Aya outperforms Qwen2-Audio on most languages, with the exception of a subset on which Qwen2-Audio is trained (i.e., English, Italian, Korean, Japanese and Chinese).

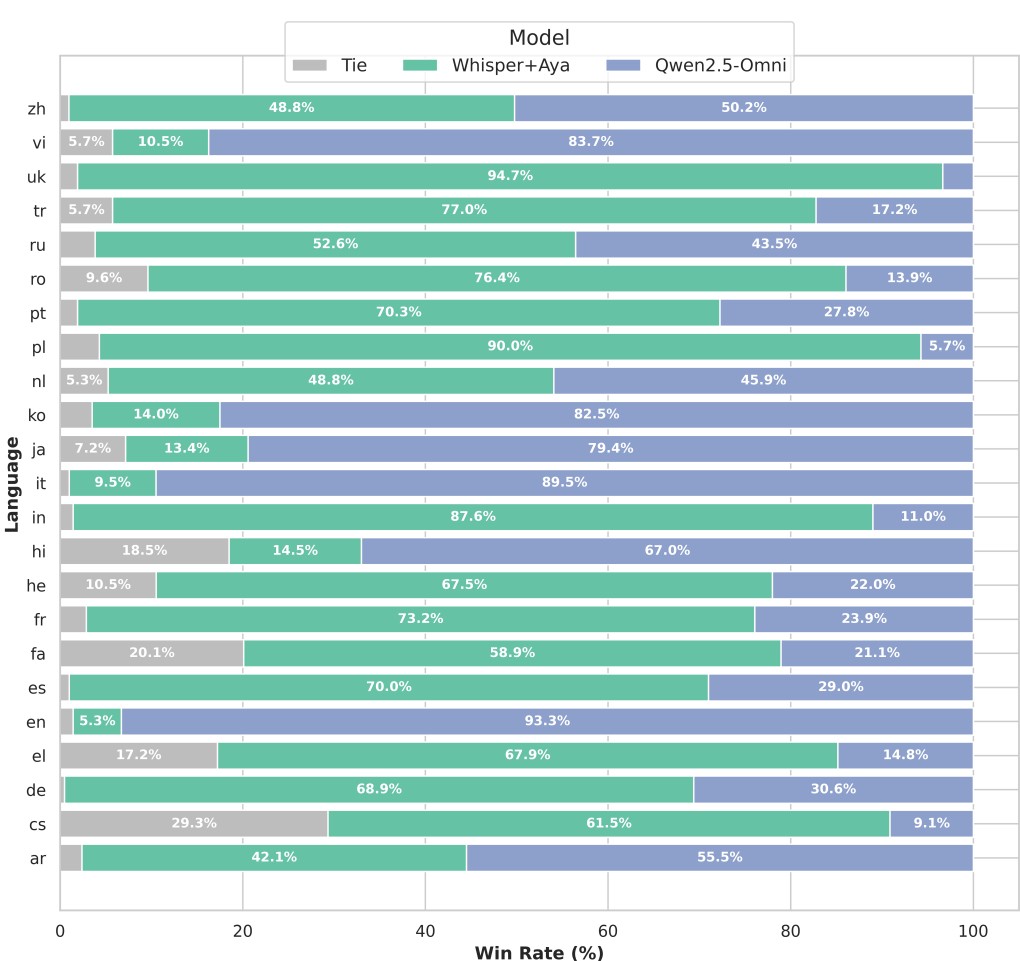

Figure 9: Win rates comparison: Whisper + Aya vs. Qwen2.5-Omni. Whisper + Aya outperforms Qwen2.5-Omni on most languages, with the exception of several Asian languages and English.

## A.2 DETAILS ON HUMAN EVALUATIONS OF SYNTHESISED QUESTIONS

We ask native speakers of the 22 non-English language to assess the naturalness and amount of content understood of a subset of synthesised questions in the test set. The participants rate the naturalness and content understood on a scale of 1 to 5.

**Naturalness:** Listen to this speech sample, then rate the naturalness of the speech.

**Content Understood:** How much of the content of the speech sample can you understand?

## A.3 WIN RATES OF QWEN2.5-OMNI VS QWEN2.5-OMNI FINETUNED

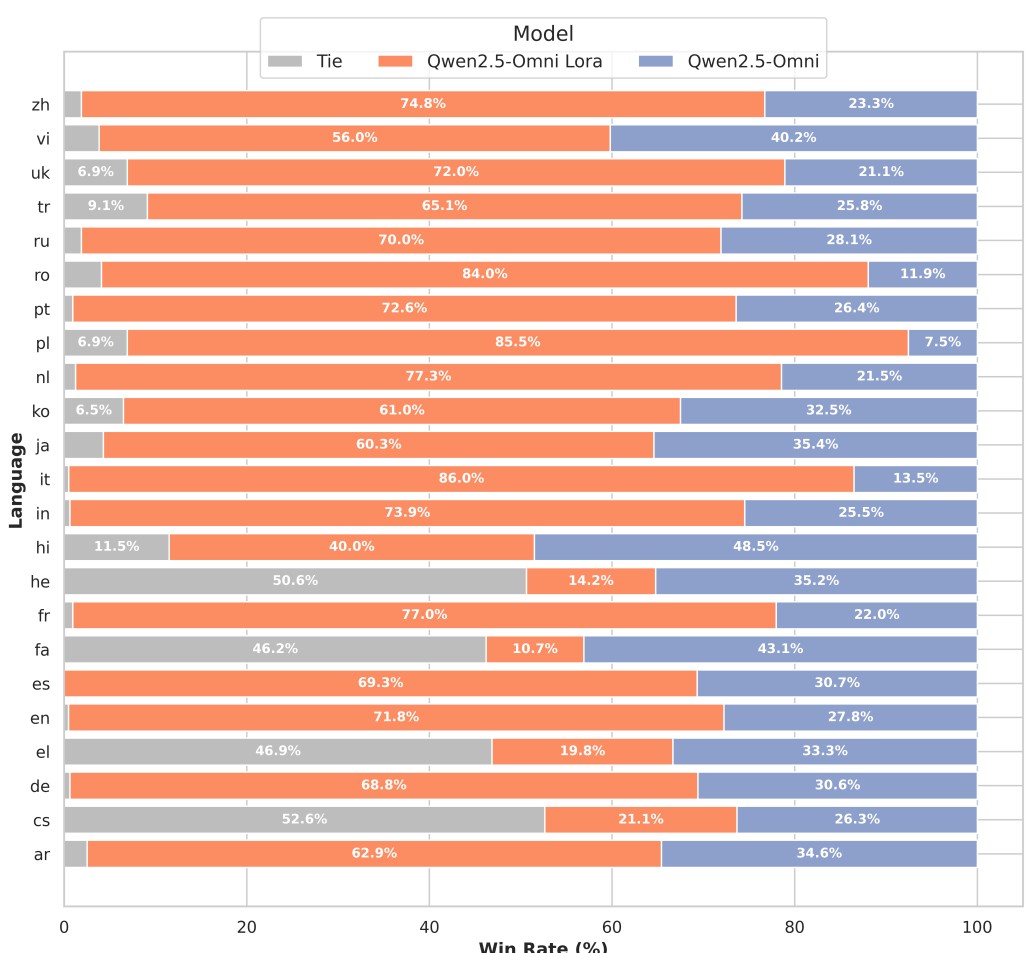

Figure 10: Win rates comparison: Qwen2.5-Omni vs. Qwen2.5-Omni finetuned. Finetuning improves performance across most languages, meaning that our MULTISPEECHQA enables better SQA capability. Hebrew, and Czech seem to lag behind, with most of the results being a tie.

## A.4 MULTISPEECH-BENCH PERFORMANCE ON QWEN2.5-OMNI MODELS

Table 4: Speech Recognition (ASR) and Speech Translation (AST) performance across models and languages: Qwen2.5 Omni vs. Qwen2.5 Omni finetuned (FT). We report CER for Chinese (zh) and Japanese (ja), and report chrF in addition to BLEU for AST performance.

| Lang. | ASR (WER %) ↓ | | AST (BLEU) ↑ | |
|---|---|---|---|---|
| | Q2.5 Omni | Q2.5 Omni FT | Q2.5 Omni | Q2.5 Omni FT |
| ar | 45.7 | 31.5 | 30.6 | 29.7 |
| cs | 100.4 | 101.6 | – | – |
| de | 7.1 | 7.5 | – | – |
| el | 108.0 | 108.3 | – | – |
| en | 13.8 | 16.6 | – | – |
| es | 4.9 | 5.1 | – | – |
| fa | 109.1 | 107.5 | – | – |
| fr | 10.9 | 10.8 | – | – |
| he | 122.9 | 113.6 | – | – |
| hi | 68.9 | 68.4 | – | – |
| id | 14.0 | 13.9 | 37.0 | 34.7 |
| it | 6.5 | 6.2 | – | – |
| ja | 75.9 | 31.3 | 17.8 | 17.2 |
| ko | 23.0 | 24.4 | – | – |
| nl | 14.0 | 14.3 | – | – |
| pl | 94.7 | 75.1 | – | – |
| pt | 9.6 | 11.9 | – | – |
| ro | 89.3 | 74.6 | – | – |
| ru | 9.3 | 13.7 | – | – |
| tr | 74.5 | 66.4 | 5.5 | 5.8 |
| uk | 82.6 | 81.3 | – | – |
| vi | 50.9 | 169.3 | – | – |
| zh | 6.2 | 6.8 | 22.7 | 17.7 |
| **Average** | 49.7 | 50.4 | 22.7 | 21.0 |

## A.5 LLM-AS-A-JUDGE PROMPT

We prompt our selected LLM with the prompt below, inserting the language name and completions for each individual pair. We randomise the answers, ensuring that each model is both Answer A or B across instances.

```
You are a helpful following assistant whose goal is to select the preferred
(least wrong) output for a given instruction in {LANGUAGE_NAME}.

Which of the following answers is the best one for given instruction
in {LANGUAGE_NAME}.
A good answer should follow these rules:
1) It should be in {LANGUAGE_NAME}
2) It should answer the request in the instruction
3) It should be factually and semantically comprehensible
4) It should be grammatically correct and fluent.

Instruction: {INSTRUCTION}
Answer (A): {COMPLETION_A}
Answer (B): {COMPLETION_B}
FIRST provide a one-sentence comparison of the two answers, explaining which
you prefer and why.
SECOND, on a new line, state only 'Answer (A)' or 'Answer (B)'
to indicate your choice.
If the both answers are equally good or bad, state 'TIE'.

Your response should use the format:
Comparison: <one-sentence comparison and explanation>
Preferred: <'Answer (A)' or 'Answer (B)' or 'TIE'>
```

## A.6 MULTISPEECH MODELS TRAINING DETAILS

We use MULTISPEECHQA to train models from scratch and glean insights on how training data mixtures affect SLMs performance. We chose the SALMONN architecture to train our models. The pretrained components are Whisper Large v3 as our speech encoder and Aya Expanse 8B as our language model. We opt for these two models based on their state-of-the-art performance in the respective multilingual capabilities. Note that the languages in MULTISPEECHQA are the same languages on which Aya Expanse 8B is trained. We use the same window length for the window-level Q-Former as in SALMONN, but replace the BERT base uncased language model encoder with an mBERT encoder to enable multilingual representations.

We hypothesise that adding speech translation data increases the task diversity in the training mixture and hence could lead a better downstream performance. For this ablation, we set a threshold of 20% for the speech translation data and use mixed batches for more robust multi-task instruction-tuning (Mueller et al., 2024).

During the training, we use parameter-efficient LoRA adapters (Hu et al., 2022) to decrease the underlying compute requirement. However, compared to SALMONN, we increase the LoRA rank and alpha to 64 for a higher trainable parameter capacity, enabling better optimization for 23 languages. We employ a multi-stage training process with different types of data for our models.

**Stage 1 ASR training** Following the SALMONN training setup, we train the window-level Q-Former and LoRA adapters using ASR data. To do this, we use a uniform amount of ASR data (20 hours) in all languages. In this stage, we add the text instruction "Transcribe this utterance" to the speech prompt. The goal of this stage is alignment between speech and text representations and enabling the model to understand the speech inputs.

We start with CommonVoice data (Ardila et al., 2020) for each language. Some of the languages have fewer than 20 hours of training data, so we balance the number of hours of data in Vietnamese with 15 hours of the Bud500 dataset (Pham et al., 2024), 18 hours of Hebrew with the Ivrit.ai dataset (Marmor et al., 2023), 14 hours Hindi with the monolingual portions of the Multilingual and Code-Switching ASR Challenges for Low Resource Indian Languages dataset (Diwan et al., 2021) and 19 hours Korean with the Zeroth-Korean corpus (Jo & Lee, 2022).

**Stage 2 Question Answering Training** For the second stage of training, we use our MULTISPEECHQA dataset for all 23 languages. For the Trivia QA, the QA Assistant, and the Alpaca GPT-4 datasets, we use all the samples, but given the difference in distribution of Anthrophic-RLHF data, we only subsample 1000 examples from this data source to ensure a training mixture that is balanced and optimized for general-purpose speech instruction-following tasks. Overall, our training mixture includes 2,070,000 samples distributed equally between 23 languages.

In addition to MULTISPEECHQA, we also run ablations where we include additional speech translation (AST) data from the CoVoST-2 (Wang et al., 2021) dataset to the training mixture.

Finally, although the model architecture allows us to append a text prompt to the speech input, we train the model without any additional text prompt to enable the question answering capability from the spoken questions alone.

| Hyperparameter | Stage 1 | Stage 2 |
|---|---|---|
| Learning rate | 1e-5 | 1e-5 |
| Warmup steps | 800 | 400 |
| LoRA Rank | 64 | 64 |
| No. of epochs | 10 | 3 |
| Batch size | 128 | 256 |
| Number of samples per language | 20 hours | 90 000 |

Table 5: Hyperparameters used to train Stage 1 and Stage 2 of MULTISPEECH models.

## A.7 HUMAN VALIDATION OF LLM-AS-A-JUDGE

In addition to checking for model calibration, we selected eight typologically diverse languages (Arabic, German, Hebrew, Hindi, Korean, Portuguese, Turkish and Chinese) to measure whether human judgements on SLM vs baseline pairs align with Command-A's judgements. Eight of 23 languages were chosen due to budget constraints. We evaluated an open model (Qwen2.5-Omni) and a commercial SLM (GPT-Audio) against the baseline and asked native speakers to judge the responses. Ensuring that we had three annotations for each pair of answers in the benchmark, we derived a consensus label from the three annotations and measured human–LLM alignment, observing 75.6% agreement for Qwen2.5-Omni ($\kappa = 0.186$) and 52.4% for GPT-Audio ($\kappa = 0.185$). For GPT-Audio, annotators showed high disagreement as the outputs are of similar quality. Overall, these experiments provide evidence that our LLM-as-a-judge setup captures human preferences to a reasonable extent, especially for Qwen2.5-Omni.

