# OpenReview forum: "Turning Speech Language Models into Multilingual Listeners"
_ICLR.cc/2026/Conference — Submitted to ICLR 2026_

### Official Review · Reviewer_YrCu · 2025-10-30

**Soundness:** 2
**Presentation:** 3
**Contribution:** 2
**Rating:** 2
**Confidence:** 5

**Summary:**

The paper tackles the scarcity of multilingual instruction-tuning data and evaluations for spoken QA by releasing MULTISPEECHQA—10.8M QA pairs (~9,200 hours) across 23 languages—built by translating VA-400K prompts/answers and synthesizing spoken questions (XTTS/Seamless/MMS), with native-speaker checks showing good comprehension but mixed naturalness. It also introduces MULTISPEECH-BENCH, which combines a human-corrected QA subset with CommonVoice (ASR) and CoVoST-2 (AST) to evaluate SQA/ASR/AST uniformly. Using LLM-as-a-judge, the authors find that among open-weight SLMs Qwen2.5-Omni is strongest on QA, while a strong cascading baseline (Whisper v3 → Aya 8B) remains competitive. LoRA-finetuning Qwen2.5-Omni on MULTISPEECHQA lifts its average QA win rate to 60.6% versus the base model, with ASR/AST staying roughly unchanged. A from-scratch SALMONN-style study shows training on all 23 languages beats using 10, while adding 20% AST data doesn’t yield consistent gains. Overall, the work argues that large-scale synthetic multilingual SQA can effectively post-train open-weight SLMs and provides datasets/benchmarks intended for community use.

**Strengths:**

1. Propose a speech dataset that has good coverage on 10.8M spoken QA pairs (~9,200 hours) across 23 languages, which is rare for SQA datasets.
2. Plans to release data and open-weight models; pipeline uses public MT/TTS. It is good for speech research community.
3.  Native-speaker ratings (≥2 raters per language, except Czech) show good content-understanding, diagnosing TTS naturalness as the main bottleneck.

**Weaknesses:**

1. QA win rates rely on a single LLM-as-a-judge (Command-A) without reported human calibration or bias checks.
2. BLEU-only for AST; lacks native-speaker evaluation and more comprehensive semantic metrics.
3. Core dataset is synthetic speech; limited evidence on robustness to real, noisy, accented speech.
4. A strong Whisper→LLM cascade remains hard to beat; ASR/AST see little improvement after fine-tuning.

**Questions:**

1. In Fig. 3 & 4, the win rates are evaluated by a single LLM (Command-A). Did you run any judge-bias checks to ensure its preferences have a positive correlation with human judgments? My suggestion is to run an experiment to verify the alignment between Command-A and human raters, rather than using Command-A simply because it covers these languages to evaluate translation quality. Otherwise, this setup may ignore the relationship between model preferences and human preferences. Thus, the resulting metric is not very convincing to me.

2. Although Qwen and Phi do not disclose their training data, they may have used real-world audio. MULTISPEECHQA uses synthetic speech—could this harm these models’ understanding of real speech? Beyond reducing compute, was LoRA also chosen to avoid such degradation? If we train a speech model from scratch and include MULTISPEECHQA as training data, would the resulting model show a gap in understanding real-world speech compared with models trained on real audio? I believe an ablation study is needed to substantiate the dataset’s quality and to assess any potential risks to speech language models pretraining.

3. Using WER for ASR is fine. But is BLEU too limited for AST? BLEU is an n-gram overlap metric and is less sensitive to paraphrases, multilingual/morphologically rich languages, and semantic equivalence. I don’t think it is a good evaluation metric. Would human evaluation by native speakers in these languages be better?

---

> ### Author Response · Authors · 2025-11-22
>
> **LLM-as-a-Judge**
>
> We agree that adding human validation to LLM-as-a-Judge is a great idea. We are now running experiments with human annotators and will report them in the coming days. We also agree that having a second LLM-as-a-Judge will help calibrate the judgements. We have calibrated the judgments with another LLM (o1) for a subset of languages and find no change, but will update the full results in our updated draft in the coming days.
>
>
> **Use of real-world speech/audio**
>
> We agree that a completely synthetic QA dataset has the chance to affect the models’ comprehension of real speech. We should note that we keep the speech encoders frozen and only use LoRA on the language model, to prevent not only degradation of real-speech understanding, but also to prevent degradation of language model capabilities. For AST, we use CoVoST-2 data, which is CommonVoice data (real speech) and find that finetuning our model does not have negative effects on real speech.
> For the proposed ablation study, would you like us to train a Whisper-style encoder-decoder speech model? We would appreciate some more detail on what you would like us to study.
>
> **Use of synthetic speech**
>
> We agree that synthetic data is not the most ideal data to use and real-world data is ideal for real-world use. However, there are cases where synthetic data has been shown to be useful. For example, in Phi-4 Multimodal, uses synthetic data to generate text QA pairs for speech prompts, generate translations and generate LM responses to speech prompts. Our work aims to extend this approach to more of the world’s languages.
>
>
> **BLEU only for AST**
>
> We agree that BLEU may be limited for AST evaluation, so we will add chrF scores to our updated draft.

---

> > ### Comment · Reviewer_YrCu · 2025-11-25
> > **Ablation Study**
> >
> > Hi,
> > I am looking forward for your next experiment results, like human validation and more evaluation metrics that provide more insights to how we can utilize this speech dataset.
> >
> > For the proposed ablation study, I think you can train either an ASR system or a TTS system. And see how robust the performance is when inference with real-world speech data (for ASR evaluation), or how reasonable the synthesized speech files are when human listen to these (for TTS evaluation).
> >
> > Thank you for your reply. Good luck!

---

### Official Review · Reviewer_zNsQ · 2025-10-31

**Soundness:** 2
**Presentation:** 2
**Contribution:** 3
**Rating:** 4
**Confidence:** 3

**Summary:**

This paper introduces MULTISPEECHQA, a large synthetic multilingual spoken QA dataset spanning 23 languages. It is created by translating the Voice Assistant 400K text pairs and synthesizing the speech with text-to-speech systems. In addition, it presents MULTISPEECH-BENCH, a multilingual and multi-task benchmark that evaluates Spoken QA, ASR, and AST tasks. The authors benchmark several open-weight spoken language models and a strong cascade baseline, then fine-tune Qwen2.5-Omni on MULTISPEECHQA, which yields higher QA win rates. Overall, the paper aims to scale multilingual instruction-following for spoken language models through synthetic data generation and standardized evaluation.

**Strengths:**

- Expanding SLMs beyond high-resource languages is both timely and impactful, as the paper addresses a key challenge: data scarcity in multilingual spoken instruction-following.
- MULTISPEECH-BENCH unifies QA, ASR, and AST tasks across the same set of languages, providing a strong cascade baseline and evaluations with multiple open-weight SLMs, which makes it a valuable resource for the community.
- The authors at least measure synthetic quality (naturalness, content) and manually verify the QA test subset, which is a good practice even if the results expose weaknesses.

**Weaknesses:**

- The heavy reliance on an LLM-as-a-judge for multilingual QA without verifying its correlation with human judgments reduces trust, especially for typologically diverse languages and speaking styles. Conducting a small human evaluation on the QA test set (beyond TTS or translation quality) would help.
- Human edits were required for 72% of translations in the benchmark subset, and the TTS outputs show only moderate naturalness on average, with notably low scores in some languages. These issues raise concerns about the overall data quality.
- Since MULTISPEECHQA and MULTISPEECH-BENCH originate from the same data source, it is not surprising that the fine-tuned Qwen model achieved state-of-the-art results on MULTISPEECH-BENCH. Moreover, the lack of a clear separation between speaker sets in the training and test data could lead to speaker overlap, which may influence the evaluation results.
- As a benchmark paper, although the authors argue that existing benchmarks lack sufficient language coverage, they do not discuss any existing speech or audio language model benchmarks. The paper would benefit from a more thorough comparison between MULTISPEECH-BENCH and other available benchmarks in this area.

**Questions:**

In addition to the weaknesses mentioned above, I have the following questions and comments:
- In Table 3, ASR results are reported using WER. However, this metric may not be suitable for certain languages such as Japanese and Chinese, where CER is typically used. Could you clarify which metric was applied?
- Typically, a table’s caption is placed above the table, but Tables 1 and 2 do not follow this convention.
- Figure 2 appears to be missing a reference in the main text.
- There are several typos throughout the paper, including “speeech,” “eplicitly,” “whcih,” “AYA Expance 8B,” and “BLUE score.”

---

> ### Author Response · Authors · 2025-11-22
>
> Thank you for your careful and considered comments and suggestions.
>
> **LLM-as-a–Judge without comparison to human judgements**
>
> We agree that adding human validation to LLM-as-a-Judge is a great idea. We are now running experiments with human annotators and will report them in the coming days. We also agree that having a second LLM-as-a-Judge will help calibrate the judgements. We have calibrated the judgments with another LLM (o1) for a subset of languages and find no change, but will update the full results in our updated draft in the coming days.
>
> **Speaker overlap in train and test set**
>
> Thank you for pointing out the possibility of speaker overlap, we agree completely! Following your suggestion, we have now recreated the test set in all languages that used voice cloning for speaker generation with new speakers. After these changes find that the finetuned model still beats the base model across those languages.
>
> **Data quality concerns**
>
> We agree that there is a variation in data quality across languages, which is a reflection of the varying TTS models available for each of the languages. Although the naturalness scores are lower, most languages have a content understood score over 4 out of 5.
>
>
> **Lack of discussion of other benchmarks**
>
> We agree that it will strengthen the paper to compare to other benchmarks, and so we are currently comparing our benchmark to AIR-Bench (as suggested by another reviewer) as well as  other benchmarks.
>
> **Presentation issues**
>
> Thank you for highlighting the various presentation issues in the paper. In fact we do report CER for Chinese and Japanese, but had not mentioned it in the draft, which we have now fixed and will upload with the new version of the paper. We have also addressed the rest of the presentation issues. We really appreciate your careful read of the paper!

---

### Official Review · Reviewer_UZCo · 2025-11-02

**Soundness:** 3
**Presentation:** 3
**Contribution:** 3
**Rating:** 6
**Confidence:** 4

**Summary:**

This paper attempts to address the lack of multilingual instruction-tuning data for Speech Language Models (SLMs), based on the fact that most existing SLMs focus on English or some other high-resource languages, limiting accessibility and global applications. It proposes a synthetic data pipeline to synthesize multilingual instruction-tuning data and open-source the synthesized datasets. The paper demonstrates that automated synthetic data pipelines can effectively scale multilingual capabilities of SLMs.

Main contributions:
1. MULTISPEECHQA - a large-scale, synthetic multilingual datasets with 10.8 million spoken question-answer pairs in 23 languages.
2. MULTISPEECH-BENCH - a multilingual benchmark suite for evaluating SLMs on SQA, ASR and AST.
3. Synthetic data pipeline for multilingual instruction-tuning employing machine translation models and multilingual TTS systems that produces high-quality speech-text pairs. The pipeline also validate the data quality by human evaluations.

**Strengths:**

Originality:
The paper introduces an approach for scaling multilingual SLMs through synthesizing and human-verifying data based off English datasets.  The originality of the paper lies in

Quality:
1. The paper covers details of the dataset construction pipeline, and introduces quantitative human evaluation by native speakers on the naturalness and comprehension quality of the synthetic data of each language.
2. The authors conducted comprehensive benchmarking on the generated evaluation set, containing cascaded system baseline and open-source SLMs such as Qwen2-Audio, Phi-4-Multimodal, etc.

Clarity
The manuscript is well-organized and easy to follow.  Proper figures and charts make the paper more intuitive.

Significant
The release of MULTISPEECHQA and MULTISPEECH-BENCH fills gaps in multilingual SLMs training and evaluation, enabling the research community to develop more inclusive and accessible SLMs.

**Weaknesses:**

Potential limitations of data quality:
1. Given the scale of the synthetic data used in MULTISPEECHQA (10.8 million samples), the scale of the human validation is 2 small (20 samples per language).
2. Variance in TTS naturalness scores may indicate that some languages contain more noise in the dataset.

Evaluation setup:
1. LLM-as-a-judge: the judge LLM's multilingual capability needs to be calibrated (perhaps using other LLMs to test the correlation/stability, or even human validation).
2. Limited task diversity in evaluation: other SLM benchmarks (such as AIR-Bench[1]) contain more diverse tasks, it may be worth extending the bench suite to cover more (e.g. summarization, language ID, emotion recognition)
3. Due to the absence of commercial SLMs (e.g. GPT4o, gemini) in benchmark, the claim of SOTA results is thin.

There's lack of ablation study, it's not clear which components of MULTISPEECHQA datasets contribute the most to the gain.


[1] Yang, Qian, et al. "Air-bench: Benchmarking large audio-language models via generative comprehension." arXiv preprint arXiv:2402.07729 (2024).

**Questions:**

See suggestions in Weakness section.

---

> ### Author Response · Authors · 2025-11-22
>
> We thank the reviewer for their comments and suggestions.
>
> **Comments on data quality**
>
> We agree that there is a variation in data quality across languages, which is a reflection of the varying TTS models available for each of the languages. Although the naturalness scores are lower, most languages have a content understood score over 4 out of 5.
>
> **LLM-as-a-Judge**
>
> We agree that adding human validation to LLM-as-a-Judge is a great idea. We are now running experiments with human annotators and will report them in the coming days. We also agree that having a second LLM-as-a-Judge will help calibrate the judgements. We have calibrated the judgements with another LLM (o1) for a subset of languages and find no change, but will update the full results in our updated draft in the coming days.
>
> **Comparison to AIR-Bench**
>
> We agree that it would be wonderful to have a  benchmark that also included many more tasks such as summarisation, language ID and emotion recognition.  Yet the focus of our work is on multilingualism, and we think it's crucial to evaluate systems on 23 languages. Creating highly multilingual benchmarks requires a lot of individual work on each language to source data and ensure it is of sufficient quality to do a task.  High-quality data for these tasks doesn't exist for the 23 languages we focus on.  So although we agree this is a wonderful future direction, it is out of scope for the current paper.
>
> **Comparison to commercial SLMs**
>
> We agree that we need to compare to commercial SLMs. We have compared to GPT-4o and will compare to Gemini and add the results in our updated draft in the coming days.

---

### Author Response · Authors · 2025-12-03
**Author response summary**

We thank the reviewers for recognizing that this work fills an important gap in multilingual speech research and will be a valuable resource for the community. We have addressed their shared concern about LLM-as-a-judge validation, along with individual suggestions on commercial models, speaker overlap, and evaluation metrics, as detailed below.

1. To validate our LLM-as-a-judge, we added an additional LLM judge and performed human annotator experiments. **After initially testing with the o1 model, we added GPT-4o as an additional LLM judge and found that both judge models give win rates within 1% of each other**, meaning that the LLMs agree with each other. For human validation, **we selected eight typologically diverse languages** (Arabic, German, Hebrew, Hindi, Korean, Portuguese, Turkish and Chinese) **to measure whether human judgements on SLM vs baseline pairs align with Command-A’s judgements**. Eight of 23 languages were chosen due to budget constraints. We evaluated an open model (Qwen2.5-Omni) and a commercial SLM (GPT-Audio) against the baseline and asked native speakers to judge the responses. Ensuring that we had three annotations for each pair of answers in the benchmark, we derived a consensus label from the three annotations and measured human–LLM alignment, observing 75.6% agreement for Qwen2.5-Omni (κ \= 0.186) and 52.4% for GPT-Audio (κ \= 0.185). For GPT-Audio, annotators showed high disagreement as the outputs are of similar quality. Overall, these experiments provide evidence that our LLM-as-a-judge setup captures human preferences to a reasonable extent, especially for Qwen2.5-Omni.
2. **As per Reviewer UZCo, we compared GPT-Audio and Gemini 2.5 Flash Lite SQA performance to the Whisper+Aya Expanse 8B baseline**. We found that GPT-Audio outperforms the cascading baseline, but Gemini 2.5 Flash Lite does not. This means that although the cascading baseline is difficult to beat for open models, some closed models already have inherent question answering ability, demonstrating that **our benchmark is a valuable contribution for measuring SQA ability of speech language models**.
3. Reviewer zNsQ pointed out that there was a potential for speaker overlap. Following this, **we recreated the test set in all languages that used voice cloning for speaker generation with new speakers**. After these changes, we found that the finetuned Qwen2.5-Omni model still outperforms the base Qwen2.5-Omni model across languages. **All the runs above were on this new dataset**.
4. Reviewer zNsQ raised presentation concerns, which we have addressed. Reviewer YrCu asked if reporting only BLEU scores for AST was too limited. Human evaluation for AST would be ideal, but is cost-prohibitive at our 23-language scale, so instead we rely on high-quality reference translations and evaluate with BLEU and chrF. We have added chrF to AST evaluation.
5. Reviewer YrCu was concerned about the use of synthetic data in our dataset. We agree that a completely synthetic QA dataset has the chance to affect the models’ comprehension of real speech. Training ASR/TTS systems as the reviewer suggested is beyond scope, but stable ASR/AST performance on CommonVoice and CoVoST-2 (real speech) after finetuning suggests that our approach preserves real-speech understanding. We note that we keep the speech encoders frozen and only use LoRA on the language model to prevent not only degradation of real-speech understanding, but also to prevent degradation of language model capabilities. As for the use of solely synthetic instruction-tuning data, there are cases where synthetic data has been shown to be useful. For example, Phi-4-Multimodal Instruct uses synthetic data to generate text QA pairs for speech prompts, generate translations and generate LM responses to speech prompts. Our work aims to extend this approach to more of the world’s languages.

---

### Meta-Review · Area_Chair_u8AS · 2026-01-03

**Summary:**

This paper proposes a large-scale synthetic data pipeline to address the scarcity of multilingual instruction-tuning data for Speech Language Models (SLMs). The main contributions is MULTISPEECHQA, a 10.8 million sample synthetic multilingual spoken QA dataset covering 23 languages. MULTISPEECHQA was built by translating VA-400K prompts/answers and synthesizing spoken questions using XTTS/Seamless/MMS, with native-speaker checks. It also introduces MULTISPEECH-BENCH, which combines a human-corrected QA subset with CommonVoice (ASR) and CoVoST-2 (AST) to evaluate Spoken QA, ASR, and AST.

Reviewers agree that releasing such a large-scale multilingual speech instruction-following corpus would be valuable to the community. The paper demonstrates that synthetic multilingual data can improve open-source SLMs via fine-tuning. Reviewers agree that the paper writing is good and easy to follow.

Overall, the work is sound. The major concerns focus on (i) synthetic data quality, and (ii) reliability of automatic evaluation with LLM-as-a-jude, especially when the major contribution of this work is dataset and evaluations. Practically speaking, it is hard to manually verify all 10.8 million samples. I will give a score of 4.

**Reviewer Concerns:**

There’s three main concerns, most other concerns have been addressed by authors during rebuttal.

**1. Reliability of LLM-as-a-judge:** All reviewers express concern that QA win rates depend heavily on a single LLM judge, without sufficient calibration against human judgments or bias/stability analysis across languages. This may weaken confidence in the reported QA improvements.


**2. Data Quality and Validation at Scale:** Human validation is limited relative to dataset size. High rates of human correction in benchmark translations and uneven TTS naturalness scores raise concerns about overall dataset noise.


**3. Related Dataset Comparison:** As mentioned by reviewer zNsQ, it would be good to compare with existing audio-language benchmarks though authors claimed insufficient coverage of languages of previous work.

The major concern is the first and the second. Authors try to address the first concern during rebuttal by adding an additional LLM judge and performed human annotator experiments. Results show that LLMs agree with each other; based on three annotators’ labeling, human–LLM agreements on the selected eight typologically diverse languages among 23 languages are as follows: 75.6% agreement for Qwen2.5-Omni (κ = 0.186) and 52.4% for GPT-Audio (κ = 0.185) — a lower reasonable level given the subjectivity of this annotation task.

**Reviewer Scores:**

4

---

### Decision · Program_Chairs · 2026-01-26

Reject